# Benefits of Best Practice Guidelines in Spine Fusion: Comparable Correction in AIS with Higher Density and Fewer Complications

**DOI:** 10.3390/healthcare11111566

**Published:** 2023-05-26

**Authors:** Pedro Fernandes, Isabel Flores, Joaquim Soares do Brito

**Affiliations:** 1Centro Hospitalar Universitário Lisboa Norte, Orthopaedics Department, Avenida Professor Egas Moniz, 1649-035 Lisboa, Portugal; 20781@chln.min-saude.pt; 2Clínica Universitária de Ortopedia, Faculdade de Medicina da Universidade de Lisboa, Avenida Professor Egas Moniz, 1649-035 Lisboa, Portugal; 3ISCTE Instituto Universitário de Lisboa, 1649-026, Lisboa, Portugal; isabel_cristina_vieira_silva@iscte-iul.pt

**Keywords:** adolescent idiopathic scoliosis, implant density, complications, outcomes

## Abstract

Background: There is significant variability in surgeons’ instrumentation patterns for adolescent idiopathic scoliosis surgery. Implant density and costs are difficult to correlate with deformity correction, safety, and quality of life measures. Materials and Methods: Two groups of postoperative adolescents were compared based on exposure to a best practice guidelines program (BPGP) introduced to decrease complications. Hybrid and stainless steel constructs were dropped, and posterior-based osteotomies, screws, and implant density were increased to 66.8 ± 12.03 vs. 57.5 ± 16.7% (*p* < 0.001). The evaluated outcomes were: initial and final correction, rate of correction loss, complications, OR returns, and SRS-22 scores (minimum two-year follow-up). Results: 34 patients were operated on before BPGP and 48 after. The samples were comparable, with the exceptions of a higher density and longer operative times after BPGP. Initial and final corrections before BPGP were 67.9° ± 22.9 and 64.6° ± 23.7; after BPGP, the corrections were 70.6° ± 17.4 and 66.5° ± 14.9 (sd). A regression analysis did not show a relation between the number of implants and postoperative correction (beta = −0.116, *p* = 0.307), final correction (beta = −0.065, *p* = 0.578), or loss of correction (beta= −0.137, *p* = 0.246). Considering screw constructs only (*n* = 63), a regression model controlled for flexibility continued to show a slight negative effect of density on initial correction (b = −0.274; *p* = 0.019). Only with major curve concavity was density relevant in initial correction (b = 0.293; *p* = 0.038), with significance at 95% not being achieved for final correction despite a similar beta (b = 0.263; *p* = 0.069). Complications and OR returns dropped from 25.6% to 4.2%. Despite this, no difference was found in SRS-22 (4.30 ± 0.432 vs. 4.42 ± 0.39; sd) or subdomain scores pre- and post-program. Findings: Although it appears counterintuitive that higher density, osteotomies, and operative time may lead to fewer complications, the study shows the value of best practice guidelines in spinal fusions. It also shows that a 66% implant density leads to better safety and efficacy, avoiding higher costs.

## 1. Introduction

The complication rate associated with pediatric spinal deformity correction surgery has been an issue of great concern and a source for multiple clinical studies. Based on essentially retrospective studies, complication rates have ranged from 5.6% to 23% for idiopathic scoliosis correction surgery [1,2,3,4,5]. Using the National Quality Forum Safe Practices (NQFSP) pediatric database, Pugely et al. [6] showed that morbidity, wound infections, reoperation rates, and readmissions for these patients were, respectively, 5.7%, 1.4%, 2.8%, and 2.7%. Since the publication in 1999 of a study that states that “to err is human”, hospitals themselves have adopted measures to improve their safety culture, investing in staff training as well as improving other factors related to surgical complications [7]. In 2012, Brooke et al., compared outcomes from hospitals that had fully implemented the National Quality Forum’s (NQF) safe practices and hospitals where this implementation had not been fully achieved yet. Despite a higher rate of complications detected, fully implemented NQF safe practice hospitals demonstrated a greater ability to diagnose and resolve problems, resulting in a lower mortality rate after high-risk surgery [8].

Technically, scoliosis surgery has progressed significantly in the last 55 years. In 1964, Harrington [9] introduced the principle of concavity distraction and correction maintained with a metal bar. Then, in the 1980s, Cotrel and Dubousset [10] introduced the concept of segmental correction by applying rod rotation, compression/distraction, and translation maneuvers, essentially fixing the spine with hooks. This methodology had an important impact on the reduction in neurological lesions and had the merit of launching the three-dimensional correction of deformity [11]. Suk et al. [12] added pedicle screws in surgical scoliosis treatment, giving surgeons the ability to apply progressively greater force in deformity reduction. Since then, several authors have shown the superiority of this type of fixation, which allows translational maneuvers and vertebral de-rotation to be performed more safely and effectively [13,14]. Another advantage, considering the capacity for correction with this type of instrumentation, is a reduced need for combined approaches to make the deformity more flexible [15]. In our experience, we have progressively replaced hybrid constructs with preferred instrumentations with screws and Ponte osteotomies [16] during posterior spinal release. The aim of this study is to evaluate the impact of a best practice guidelines program (BPGP) in idiopathic scoliosis surgery, a program based on a detailed analysis of the literature published throughout 2010 [17,18,19,20,21,22,23,24,25,26] that was implemented parallel to the use of posterior osteotomies and an increase in implant density. A particular emphasis is given to unplanned operating room returns (UORRs) and deformity correction throughout the study period.

## 2. Materials and Methods

### 2.1. Quality and Safety Measures

In the preoperative period after BPGP measures were implemented, patients were also evaluated based on their comorbidities; hemoglobin was raised above 12 g/dL with iron and erythropoietin when needed. Autologous blood donation was suspended as previous patients all had a significant decrease in hemoglobin preoperatively after donating blood. A preoperative bath was implemented with chlorohexidine both the day before surgery and the day of surgery. Alcohol-based chlorohexidine was introduced for skin prep at the time of surgery, and a time-out with a checklist was implemented. Anesthesiology introduced tranexamic acid in a dose varying between 10–30 mg/kg initially, followed by 3 to 5 mg/kg/h of perfusion. Progressive remifentanil infusion replaced the use of fentanyl during anesthesia. Stainless steel instrumentation was abandoned and replaced by titanium-with-cobalt chrome rods; posterior osteotomies in the curve apex became routine. Hybrid constructs were already being replaced by all-screw constructs with higher implant density due to surgeons’ increased familiarity with thoracic pedicle screws. Multimodal neuromonitoring was introduced, replacing somatosensory-only monitoring. In the postoperative period, a restrictive transfusion policy was initiated, thus avoiding transfusion when hemoglobin was above 7 g/dL following an uneventful surgery with no clinical signs of hemodynamic instability.

Colleagues and staff were persuaded to deviate from usual practice by following a unit protocol supervised by the lead surgeon, including time-out for the preoperative safety checklist. The multimodal approach for decreasing transfusions was also instituted and controlled by the Anesthesia and Hemotherapy Department. The number of people in the operating room was controlled by the surgeon, but keeping to the minimum numbers possible was not always accomplished and was difficult to quantify. The amount of irrigation fluid was increased to 300 cc after exposure, instrumentation, osteotomies, and rods were placed but before bone grafting and closure. All patients were operated on by the lead author (who has two years of practice following a one-year spine fellowship) with the same exposure and dissection techniques.

### 2.2. Inclusion Criteria

Patients included were under 21 years of age and had been diagnosed with idiopathic scoliosis requiring posterior instrumentation at five or more levels, regardless of previous anterior or posterior releases. Patients were operated on between 2006 and 2016, with data gathered retrospectively until 2013 and collected prospectively between 2014 and July 2016.

### 2.3. Radiographic Analysis

All deformities were evaluated via standing long anteroposterior and sagittal radiographs with the major Cobb angle measured preoperatively (CobbM), postoperatively (Cobb1), and at the patient’s last evaluation (Cobb2) with a minimum follow-up of 24 months. The percent of correction was calculated using the CobbM–Cobb1 or CobbM–Cobb2 formula divided by CobbM and multiplied by 100. We defined implant density as the ratio of the number of implants applied (hooks, screws, and wires) to the number of vertebrae instrumented, times two (Figure 1 and Figure 2). A 100% density corresponds to a density of two implants per level.

### 2.4. Patient-Reported Outcomes

Quality of life was assessed by the SRS-22 questionnaire obtained at the patient’s last visit, with a minimum follow-up of 24 months. Results were compared between the two groups.

### 2.5. Statistical Analysis

Samples were characterized using means for age, Cobb angle, levels fused, and follow-up. For comparison analysis, all variables were tested for the normality of the distribution, and since this was not always assured, a non-parametric Mann–Whitney U test (two independent samples) was used with alpha set at 0.05. The median values and respective interquartile ranges were reported. The independent factor was surgery performed before or after BPGP implementation. The effect size was calculated using Cohen’s d (when normality was assumed) or eta-squared (for non-normal distributions). We considered effect sizes of at least 0.2 to reflect a clinically relevant effect of the BPGP. For categorical variables, a chi-squared test was used with effect size given by odds ratio (OR) adjusted to post-program, where 1 was coded for post-program to obtain Cochran–Mantel–Haenszel statistics. Through ANCOVA analysis and logistic regression, correlations between implant density and initial and final correction were calculated, as well as loss of correction, taking into consideration exposure to the BPGP and curve flexibility.

## 3. Results

### 3.1. Sample

We studied 82 adolescents (69 females) operated upon consecutively for idiopathic scoliosis via a posterior approach. Sixty patients had a thoracic curve and twenty-two had a thoracolumbar curve. The mean age was 14.91 ± 2.6 years (range 10 to 21), and the mean Cobb angle was 62.9° ± 15.85° (range 35° to 114°). The average follow-up was 48.9 months (with a range of 24 to 124) for the two groups combined. On average, 11.6 ± 2.27 levels were instrumented (with a range of 5 to 15). Additional procedures for anterior curve release were performed in eight patients. Hybrid constructs were used in 19 cases, but screw constructs were used predominantly, in 63 patients. Selective fusions were performed for 31 patients and thoracoplasties for 39 patients.

### 3.2. Two Samples Comparative Analysis

Thirty-four patients (76.5% female) were operated on in the pre-BPGP period and 48 patients (89.6% female) after BPGP implementation. The two populations were comparable with respect to age, curve severity and flexibility, sagittal profile, and the number of levels fused. In the pre-BPGP period, anterior releases were performed in three cases (8.8%); after BPGP, a complementary anterior or posterior release was performed in five cases (10.9%). The implant density increased after BPGP with 66.8% ± 12.03, compared with 57% ± 16.7 implant density before BPGP (*p* < 0.002; Cohen’s d = −0.114). This increase was also relevant in major curve density on both the convex and concave sides. The operative time increased after the implementation of quality measures (*p* < 0.012; Cohen’s d = −0.51) (Table 1 and Table 2).

Before BPGP, the initial median correction was 67.9° ± 23, while after BPGP, it was 70.6° ± 17.4 (*p* = 0.147; Cohen’s d = 0.16). The median final correction was 64.6° ± 23.7 pre-BPGP and 66.5° ± 14.9 post-BPGP (*p* = 0.078; Cohen’s d = 0.19). The loss of reduction, measured as a ratio of initial correction (Correction 1) over final correction (Correction 2), was 1.075 ± 0.14 in the pre-BPGP period and 1.017 ± 0.14 after the BPGP (*p* = 0.143; Cohen’s d = 0.16) (Table 3).

### 3.3. Major Complications (UORR)

With a minimum follow-up of six years, eleven patients returned to the OR for reoperation; before BPGP, there were six late infections, one proximal junctional failure, one distal add-on, and one secondary thoracoplasty. In the post-BPGP period, there was one early wound dehiscence and one late infection, each requiring an unplanned surgical procedure (Table 3). Regarding late infection, patients were re-operated on after a mean period of 33 months (min: 12, max: 96), with isolation of the bacterial agent in four patients. In six of these cases, stainless steel material was used, and in only one case, titanium was used. Most patients presented with a purulent collection around the implants and, frequently, with peri-implant corrosion. All patients were treated by means of material removal and antibiotic therapy. The junctional failure case occurred after hybrid instrumentation.

### 3.4. Quality of Life

As can be seen in Table 4, no difference was found in the SRS-22 scores between the two populations, showing that the measures introduced had no relevant impact on patients’ overall quality of life (d = 0.29; *p* = 0.386).

### 3.5. Cost Analysis

After BPGP implementation, the cost of implants increased from EUR 7584.26 (range: 7163.90–8004.63) to a mean of EUR 8571.67 (range: 8128.98–9014.36) (beta = 0.333, *p* = 0.002). A logistic regression analysis showed that the difference decreased to EUR 600 when the cost was controlled for levels and density. Considering an average of 11.6 levels, the model predicted an increase of EUR 1915 per patient when the density from the mean increased from 65.4% to 100%.

### 3.6. Curve Correction

In an ANCOVA analysis, using the initial correction as the covariate and the exposure to osteotomies and screws and the increased implant density as discrimination variables, we showed that this approach was not a good predictor of the final correction: only the initial correction was correlated with the final correction. Although the model was relevant (F (1.77) = 4.723, *p* = 0.033), the difference between variables revealed that the effect happened only in relation to the initial correction. Whether surgery was performed before or after quality measures did not affect the final correction (F (1.75) = 1.142, *p* = 0.289). As can be seen in Figure 3, the only borderline effect on the final correction was the lessened occurrence of smaller corrections after the BPGP.

In order to know if the correction of a deformity was in any way correlated with the use of more or fewer implants (density), linear regressions were performed in two modules in which the percentage of correction was sought for the number of implants used for each and also controlled for the timing of surgeries (Figure 4 and Figure 5). The regression analysis clearly showed that the number of implants was not related to postoperative corrections (beta = −0.116, *p* = 0.307) or to the final corrections (beta = −0.065, *p* = 0.578). Finally, in an attempt to understand if implant density decreased correction loss, the ratio of Correction 1/Correction 2 was used, meaning that when this was greater than one, a loss of correction or progression of the deformity was documented. After removing an outlier that had almost complete loss of correction, the regression analysis showed that the number of implants was not related to correction loss (beta = −0.137, *p* = 0.246). Although no correlation was found between the density and ratio (Correction 1/Correction 2), Figure 6 shows a greater number of patients falling below the reference line, meaning a lower loss of correction after the BPGP period. A linear regression model that included only all-screw construct patients (*n* = 63) was performed to evaluate the impact of overall density, major curve convexity, and concave density on curve correction (Table 5).

The initial correction variance can be partially explained by the concave major curve density (beta = 0.293 and *p* = 0.038), while the convex density was correlated with a smaller loss of correction (beta = −0.3 and *p* = 0.027) when the ratio of correction was used as the dependent measure. In the same model, the overall density had a negative effect on the initial correction (b = −0.393 and *p* = 0.01). Figure 7 shows how the concave implant density and flexibility correlate with the estimated correction, where we can see that the same level of correction obtained in a flexible curve with a low-density construct can only be obtained in a rigid curve with a high-density construct.

## 4. Discussion

Idiopathic scoliosis is a complex deformity of the spine, and the extent of any surgical procedure for correction depends on the size of the curve, its flexibility, the extent of release, and the number and type of anchors used, whether hooks, wires, or screws. Despite technical advances and developments in anesthesia, surgical techniques, and intensive care support, which all enable the performance of increasingly complex surgeries, the surgical correction of scoliosis still requires long operative times, extensive approaches, and the use of a considerable number of implants. During the year 2011, our previously unsatisfactory outcomes led us to adopt a program of quality and safety measures at a full-institutional level, during the preoperative, intraoperative, and postoperative periods, aiming to minimize adverse events and complications in pediatric spinal deformity surgery. The number of unplanned returns to the operating room is one of the most relevant parameters to be considered, usually representing a major complication. Insufficient correction, decompensation, fusion problems (crankshaft, fixation failures, and pseudarthrosis), and infection are the most frequent causes, as are transitional deformities such as adding-on or junctional kyphosis. Several return-to-the-operating-room rates can be found in the literature, though these rates are greatly dependent on the length of the follow-up period [27,28,29]. Haynes et al., in 2009, were able to clearly show the efficacy of these measures by presenting the results of the checklist policy implementation in various geographical areas and with varying economic realities, with a decline in mortality (1.5 to 0.7%; *p* < 0.01) and in complications during hospitalization (11% to 7%; *p* < 0.01) [19]. The significant decrease in UORRs in our series, mostly by avoiding late infection, even taking into account longer and more complex procedures, is a good example of the value of safety measures in complex surgery. Apart from infection, junctional failure explained one of our UORRs, and this complication can be related to the surgeon’s experience and the type of implants used. In a series by Samdani et al., the return-to-the-operating-room rate with hybrid fixation systems, namely proximal hooks, as in our case, was 12.5%, whereas, for the group with screws, the rate was only 3.5% (*p* < 0.01) [30]. The same was reported by Kuklo et al., when they compared instrumentations with hooks, hybrid instrumentations, and all-screw instrumentations, thus demonstrating the superiority of the latter [31].

Suk et al. [12] compared the rates of correction among constructs with hooks, screws arranged in a hook configuration, and screws only for all segments. Deformity corrections, respectively, were 55%, 66%, and 72%, with correction losses of 6%, 2%, and 1%. According to these authors, the use of screws gave better results in sagittal plane corrections, vertebral de-rotations, and deformity control [16]. Lee et al. [32] had already demonstrated the efficacy of apical de-rotation maneuvers with the use of screws, obtaining a decrease in vertebral rotation of 42% and a major curve correction of 79%. Kim et al. [13], when comparing hybrid instrumentation with all-screw constructs, could not verify a significant difference in the correction percentage with screws. However, a lower correction loss was observed with screws, with apical vertebral rotation only slightly corrected in both instrumentations. Kuklo et al. [31] and Rose et al. [33] demonstrated a lower revision rate for instrumentation failure with the use of pedicular constructs.

Most studies, although showing evidence for the superiority of screw fixation, fail to randomize patients for spinal flexibility beyond any curve severity [31,33]. In this context, Vora et al. [34] and Rose et al. [33], when controlling for the flexibility of curves, by comparison, could not demonstrate a clear superiority between hybrid and screw instrumentations. Larson et al., and the Minimize Implants/Maximize Outcomes study group (MIMO) presented an extensive literature review, selecting 10 studies from 196 references where implant density, calculated differently than in our study, varied between 1.06 and 2.0 implants per level, with the mean curve correction varying between 64% and 70%. Great heterogeneity in anchor density was verified, with most studies being too underpowered to draw out any evidence on the relationship between the number of implants and outcomes in adolescent idiopathic scoliosis [35]. Subsequently, the same group studied 952 adolescents operated on with predominantly screw constructs, divided between a low density (<1.54) and a high density (>1.54) per level and controlling for curve flexibility and fusion length. They were able to find a statistically significant difference in curve correction from a mean of 66% to 69%. The same was also found for SRS-22 total scores and function, appearance, and satisfaction domains [36]. According to the authors, only function (with a difference of 0.11) reached the minimal important clinical difference (MICD) defined by Carreon et al. [37]. This, however, remained well below the MICD value calculated using the error-based method introduced by Bago et al. [38], leading us to question the real impact of this difference on the outcomes of AIS surgery.

Rushton et al. [39] studied the economic costs of scoliosis surgery, questioning whether the differences in outcomes compensated for the financial burden of screw constructs. While, in some cases, the advantage of screws is undeniable, especially in patients with myelomeningocele or in three-column osteotomies, in other cases, this advantage may not be so obvious. This is especially true when the surgeon may contemplate hybrid instrumentation or a configuration with screws but with less implant density. Sanders et al. [40] also highlighted the lack of consensus among surgeons regarding the type of instrumentation, levels to be treated, and type of approach. The implant cost for a typical thoracic curve surgery ranged from USD 22,824 to 40,992, depending on the use of wires, hooks, or the more costly instrumentation with screws. According to Rushton et al. [39], implants account for 20–30% of the total cost of surgery, and this may approach 50% when the implant density exceeds 80%. The MIMO study group estimated the cost difference between a high-density group (1.8 implants per level) and a standard-density approach (1.48). They also included the anticipated rate of screw malpositioning and the costs associated with anticipated revisions. By using the standard density, meaning <3.2 screws per patient, and by avoiding revisions for screw malpositioning (rate 0.12–0.48%), a 7% reduction in national hospitalization costs could be attained [36]. Although the predictive cost analysis included the use of predominantly screw constructs (but also hooks and a few sublaminar wires), our cost analysis showed that one patient could have a 65.4% density construct for free out of every four patients operated on with 100% density. Considering our low complication rate after BPGP, related also to avoiding stainless steel and our correction rate, compared well with what is currently published in the literature [35], we can state that efficacy and safety can be achieved with a 66.8% density. The relevance of choosing less expensive constructs is also well documented by Cheng et al. who compared clinical results in a population with idiopathic scoliosis where sublaminar wires were used in the apical region instead of pedicular screws. The fixation levels were identical, the correction and correction loss rates overlapped, and the clinical outcomes were similar, according to the SRS-22 questionnaire [41]. Considering the excellent results with less costly instrumentation, it becomes essential to look for evidence that may or may not justify the use of screws in general and especially in constructs with a 100% density [42].

In our series of patients, we did not find significant radiographic differences between the two studied periods, before and after BPGP, when the rates of initial and final correction and correction losses at follow-up were compared. Another concern in our study was the relationship between implant density and the rate of correction loss at follow-up. Rushton et al. [39], in a multicenter study, also failed to establish a correlation between implant density and deformity correction. On the other hand, the planned instrumentation had little to do with deformity data, being independent of curve size and with considerable variability in the choice of levels, implant distribution, and respective implant density. Our results lead us to believe that more pedicular instrumentation and the introduction of posterior osteotomies, mostly used after the BPGP, had a borderline impact on the radiographic and self-reported outcomes evaluated in a binary way. Only concave major curve density (upper to lower major curve vertebrae) was correlated with improved curve correction, a finding similarly documented (to some degree) by Le Naveaux, where apical concave implant density (apex ± one vertebra) had a positive effect on curve correction [43]. We also found that convex major curve density was correlated with a lower loss of correction in a model where overall density had a relevant negative impact on the initial major curve correction. This may well explain the difficulties in showing a correlation between implant density given as an overall measurement when looking for its impact exclusively on the major curve [35,36]. Finally, we could not demonstrate a real gain in the overall quality of life with the BPGP, as SRS-22 scores, as well as the different domains’ responses, showed no significant difference between the two studied periods. This was also stated by different authors [44,45,46,47,48,49], where no correlation was shown between quality of life and radiographic parameters.

In line with what has been stated, different authors all aim for safer surgery with less operating time and lower blood loss. They have presented their results for the skip pedicle configuration [50,51] and for the major convex curve side pedicle instrumentation, with similar curve correction rates but fewer screws and a low complication rate [52].

Although with a long follow-up, our study has several limitations because the data were collected over a ten-year period with the potentially confounding variables of technological advances, improved surgical techniques, and lessened surgeons’ learning curves. Both the choice of levels to be instrumented and the pedicle screw fixation of the thoracic spine are skills that are progressively developed during any surgical practice. Cahill et al., compared surgical outcomes in adolescent idiopathic scoliosis (AIS) between younger versus more experienced surgeons and found differences only in estimated blood loss (EBL), surgical time, and SRS scores, but not in complication rates [53]. The same author studied differences in outcomes in neuromuscular patients, again between younger surgeons and those with more experience, but the outcomes differed only in operative time and the number of fused segments [54]. Other results were similar. Lonner et al., analyzed their learning curve for performing AIS thoracoscopic correction and were able to demonstrate a reduction in operative time and Cobb correction toward the second half of all patients operated on, but complications were equally distributed along the whole sample [55].

## 5. Conclusions

Given the complexity of idiopathic scoliosis, where several factors need to be addressed, compliance with a safety culture is paramount to improving outcomes, mainly by avoiding complications. On the other hand, it will be very difficult to develop guidelines for instrument configuration and density. However, given the progressive increases in the cost of treatment and the present lack of good evidence on any relevant benefits of constructs with a high density of pedicle screws, it will continue to be the responsibility of the surgeon to choose the most appropriate number of implants based on the best cost/benefit/security ratio.

## Figures and Tables

**Figure 1 healthcare-11-01566-f001:**
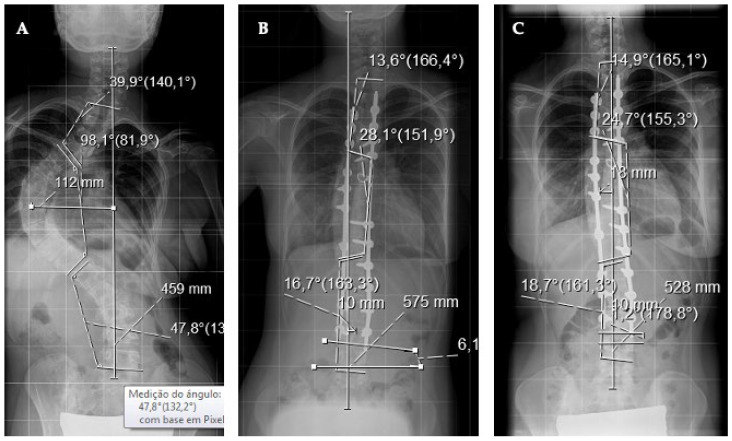
(**A**) Anterior–posterior radiograph of adolescent idiopathic scoliosis, Lenke II NA, with major thoracic curve of 98 degrees. (**B**) Postoperative radiography showing posterior instrumentation from T3–L3 with correction of the deformity to 28 degrees (a 71% correction). (**C**) Correction at 24 months with curve of 24.7 degrees, within the margin of error (4–6 degrees), showing no correction loss. Twenty implants were used at 26 possible points of fixation given the 13 instrumented levels (density: 76%).

**Figure 2 healthcare-11-01566-f002:**
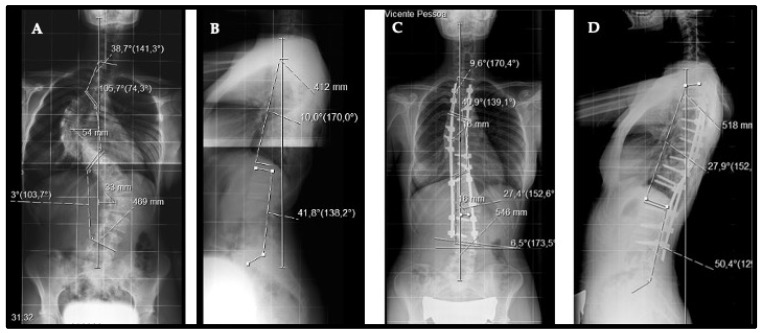
(**A**,**B**) Anterior–posterior and lateral radiograph of a 15-year-old adolescent with a Lenke IV (−) C with major thoracic curve of 105 degrees. (**C**,**D**) T2–L3 posterior spinal fusion was performed using a 65% implant density construct with the correction shown at 2 years.

**Figure 3 healthcare-11-01566-f003:**
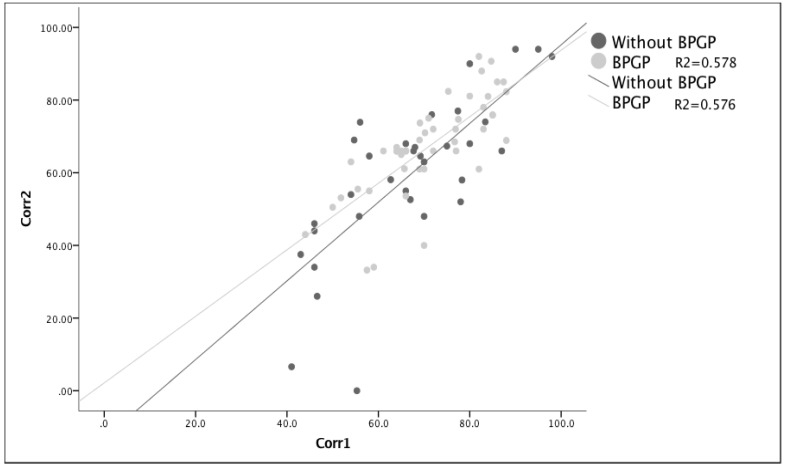
Visualization of the covariate effect of the final correction (Corr 2) and initial correction (Corr 1) through surgery before and after exposure to quality measures (BPGP). The final correction is positively linked (r = 0.76) to the correction obtained after surgery, and the quality of the relationship is the same before and after BPGP. This is considered a big correlation, meaning that, on average, for one point in correction right after surgery, a final correction of 0.76 can be expected.

**Figure 4 healthcare-11-01566-f004:**
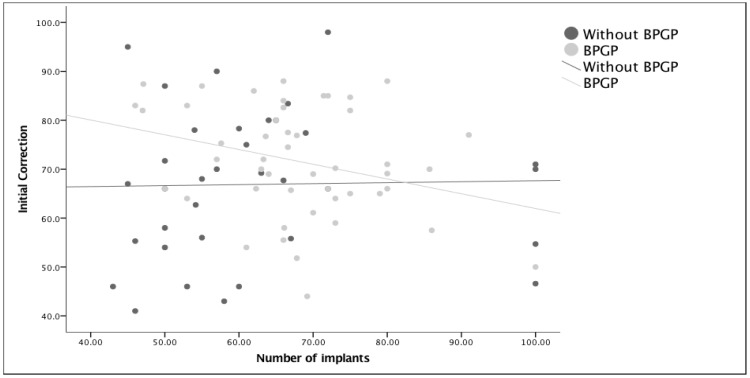
Visualization of the covariate effect of the initial correction (Corr1) on the number of implants per surgery before and after exposure to quality measures (BPSP). Density and percentage of correction after surgery (R2) are very small for both relationships (R2 (without QSM) = 0.001 and R2 (with QSM) = 0.09)) meaning that post-surgical correction was not linked to the number of implants. After quality measures (BPGP), there was a slight negative correlation with a decrease in correction in patients with a higher number of implants, though this relationship was very minor (R2 = 0.09).

**Figure 5 healthcare-11-01566-f005:**
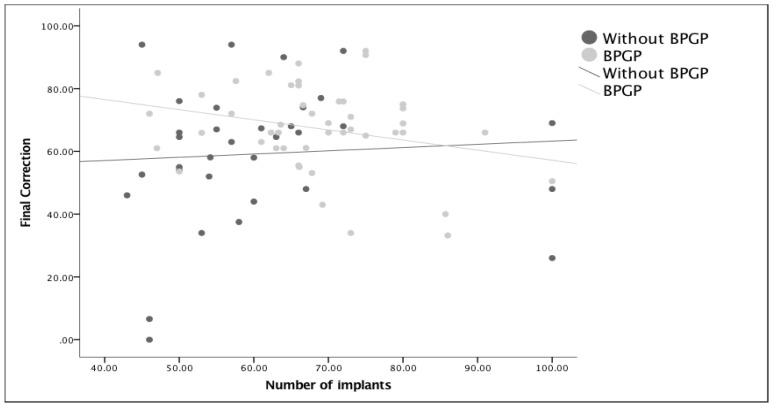
Visualization of the covariate effect of the final correction (Corr2) on the number of implants per surgery before and after exposure to quality measures (BPGP). Density and percentage of correction after surgery (R2) are very small for both relationships (R2 (without BPGP) = 0.005; R2 (with BPGP) = 0.072), meaning that final correction was not linked to the number of implants. After quality measures (BPGP), there was a slight negative correlation with a decrease in correction in patients with a higher number of implants, though this relationship was very minor (R2 = 0.072).

**Figure 6 healthcare-11-01566-f006:**
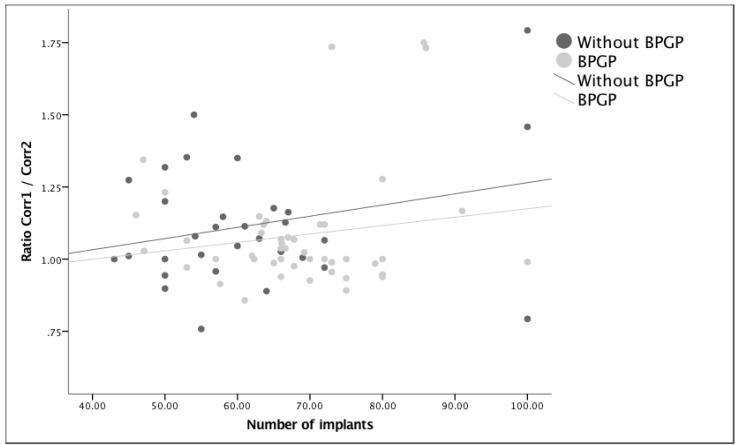
The relationship between density and loss ratios, excluding outliers (Correction 1/Correction 2). A positive ratio represents loss of correction as Correction 1 is greater than Correction 2. This is shown before and after exposure to BPGP.

**Figure 7 healthcare-11-01566-f007:**
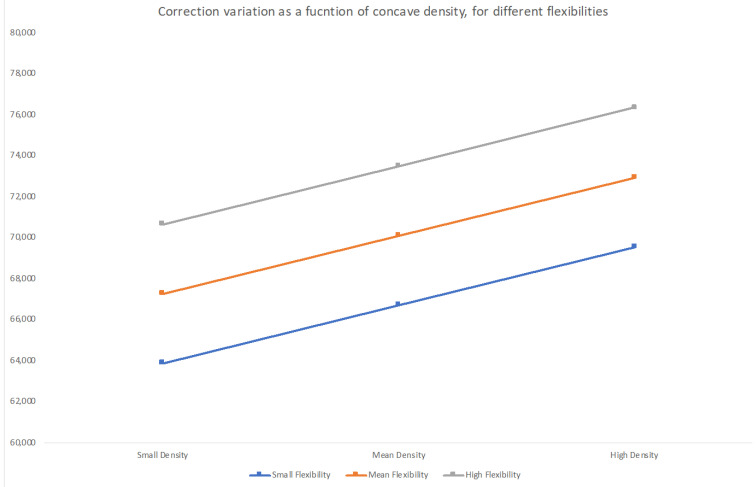
Model estimate for postoperative correction (Correction 1) based on implant density and curve flexibility. A positive correlation was only seen in the concave density model. There, we can see that to obtain (in a rigid curve (−1 quartile; lower curve)) the same correction as in a flexible curve ((+1 quartile; upper curve) being treated with fewer implants (−1 quartile)) we will need to use the highest density pattern (+1 quartile).

**Table 1 healthcare-11-01566-t001:** Population characteristics before/after (BPGP).

Characteristic	Pre-BPGP (34) Median ± IQ	Pos-BPGP (48) Median ± IQ	Mann–Whitney U *p* Value	Eta-Squared (η)
Age (Years)	14 ± 3.3	14 ± 2.8	0.59	0.004
Pre-op Hb	12.5±1.8	13.8±1.3	0.01	0.11
Risser	4 ± 2	4 ± 1	0.343	0.011
Major Cobb angle	60° ± 17.3	60° ± 18.6	0.42	0.0081
Major Flexibility (%)	36 ± 23.4	36.3 ± 22.5	0.862	0.001
Levels Fused (n)	12 ±2.3	12 ± 3.0	0.98	0.001
Operative Time (m)	260.46 ± 45.4	286.72 ± 57.5	0.012	0.51
Total density (%)	57.5 ± 16.7	66.8 ± 12.03	0.002	0.114
Density MC (%)	50 ± 19.05	66.6 ± 20.3	<0.001	0.22
Convex_Density (%)	50 ± 23.8	59.57 ± 35.0	0.027	0.06
Concave_Density (%)	50 ± 24	71.2 ± 28.1	<0.001	0.25
Density/Density MC	1.22 ± 0	1.01 ± 0	0.001	0.15
Cost (EUR)	7700 ± 2025	8750 ± 2192.50	0.003	0.11

Continuous variables based on median scores and quartiles and comparisons made with Mann–Whitney statistics and eta-squared calculation for effect dimension. (MC: major curve; BPGP: best practice guidelines program).

**Table 2 healthcare-11-01566-t002:** Population characteristics before/after (BPGP).

Characteristic	Pre-BPGP (34)	Pos-BPGP (48)	Chi-Squared Difference (Sig.)	Effect (Odds Ratio)
Gender (F)	77%	90%	0.11	2.646
Additional Procedures (Yes)	8.8%	10.4%	0.811	1.2
Selective (Yes)	44.1%	33.3%	0.312	0.633
Hybrid (Yes)	33.3%	2.1%	<0.000	0.043
Thoracoplasty (Yes)	55.9%	41.7%	0.20	0.564
Lenke 1, 2 (N)	63.6%	66.7%	0.647	1.238
Lenke 3, 4 (N)	15.2%	12.5%	0.773	0.829
Lenke 5, 6 (N)	21.2%	20.8%	0.978	0.015
Modificator A	36.4%	27.1%	0.426	0.681
Modificator B	6.1%	10.4%	0.469	1.86
Modificator C	55.9%	62.5%	0.547	1.316

Categorical variables are presented as percentages and comparison significance made with chi-squared statistics and effect size given by odds ratio. (Additional procedures: anterior or posterior releases before posterior fusion.)

**Table 3 healthcare-11-01566-t003:** Percent of correction, complications, and returns to operating room by group.

Data	Pre-BPGP (34)	Post-BPGP (48)	*p* Value	Cohen’s D
Immediate (%)	67.85 (23.20)	70.10 (17.40)	0.15	0.16
Final (%)	64.60 (23.68)	66.50 (14.90)	0.08	0.19
Ratio (Immediate/Final)	1.08 (0.14)	1.02 (0.14)	0.14	0.16
Complications (%)	26.5 (9)	4.2 (2)	0.01	0.12
Returns to OR (%)	26.5 (9)	4.2 (2)	0.01	0.12

BPGP: best practice guidelines program; OR: operating room.

**Table 4 healthcare-11-01566-t004:** Clinical outcome before and after BPGP.

Score	Pre-BPGP	Post-BPGP	S (*p*)	Cohen’s D
SRS-22 total	4.37 ± 1.92	4.52 ± 0.49	0.114	0.002
Function	4.2 ± 0.5	4.2 ± 0.6	0.657	0
Pain	4.4 ± 0.8	4.6 ± 0.8	0.122	0.25
Self-image	4.45 ± 1	4.4 ± 0.6	0.296	0.002
Mental health	4.2 ± 0.8	4.3 ± 1	0.263	0.11
Satisfaction	5 ± 0.5	5 ± 0	0.126	0

BPGP: best practice guidelines program.

**Table 5 healthcare-11-01566-t005:** Correlation between correction and global density and density over the major curve (end vertebrae to end vertebrae given by convexity and concave density).

Data	Corr 1		Corr 2		Ration Corr 1/Corr 2	
	Beta	*p* Value	Beta	*p* Value	Beta	*p* Value
Density	−0.393	0.01	−0.180	0.237	0.017	0.911
Convex_D	0.017	0.899	−0.138	0.306	−0.3	0.027
Concave_D	0.293	0.038	0.263	0.069	−0.011	0.937

Major curve density, as it was strongly related to convexity and concave density, was removed from the analysis. Due to the small sample (*n* = 63), flexibility was not introduced in the model.

## Data Availability

The data presented in this study are available on request.

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
