# Peer review of "Benefits of Best Practice Guidelines in Spine Fusion: Comparable Correction in AIS with Higher Density and Fewer Complications"

_healthcare, 2023, doi:10.3390/healthcare11111566_

Round 1
Reviewer 1 Report
Dear Authors
Congratulations on the study. I have some comments that need clarification before considering your work for publication.
1. The title is misleading since apart from many other factors that were implemented and evaluated why to focus on the implant density alone in the title instead it could be revised to "Impact of Implementation of Safety Protocol in Adolescent Idiopathic Scoliosis Surgery"
If the safety protocol has recommended or been implemented with only an increase in the implant density then it should hold good both for the study design and the title but there are an array of things that were modified from the routine protocol hence just highlighting the implant density does not sound good
Hence change the aim of the study alone to evaluate the impact of the safety protocol and not only the implant since its an overall estimate and not a point estimate of its effect
Explain the DIM effect in the statistical section
Link the figures to the results and not to present them as a standalone entity to make it easy for the readers to follow the context presented
Limitations of the study was not highlighted
The conclusion could be separately made out
Author Response
- “The title is misleading since apart from many other factors that were implemented and evaluated why to focus on the implant density alone in the title instead it could be revised to "Impact of Implementation of Safety Protocol in Adolescent Idiopathic Scoliosis Surgery."
Reply: We have debated this issue and we truly agree with the comment. This manuscript was presented as a podium presentation at the SRS annual meeting with the title “Benefits of Best Practice Guidelines in Spine Fusion: Comparable Correction with Higher Density and Fewer Complications”. Our suggestion would be change for Benefits of Best Practice Guidelines in Spinal Fusion: Comparable Correction in AIS with Higher Density and Fewer Complications. In fact, Implant density increase was only part of a full change in practice with safety guidelines being fully implemented.
- “If the safety protocol has recommended or been implemented with only an increase in the implant density, then it should hold good both for the study design and the title but there is an array of things that were modified from the routine protocol hence just highlighting the implant density does not sound good.”
Reply: We do agree.
- “Hence change the aim of the study alone to evaluate the impact of the safety protocol and not only the implant since it’s an overall estimate and not a point estimate of its effect.”
Reply: Done and applied within the final manuscript.
- “Explain the DIM effect in the statistical section.”
Reply: In the case it is Cohen d – correction is introduced. This analysis was performed because we had a relatively small sample. Odds ratio was used for categorical variables and eta test for continuous variables.
- “Link the figures to the results and not to present them as a standalone entity to make it easy for the readers to follow the context presented.”
Reply: We introduced a second case for the results as the pictures are in M&M just because the protocol is discussed
- “Limitations of the study was not highlighted.”
Reply: They were introduced within the final manuscript
- “The conclusion could be separately made out.”
Reply: It is done in the final manuscript
Reviewer 2 Report
Dear Authors, I would like to congratulate you on your study. The modalities of the study are clearly illustrated and the evaluation methods are adeguate. However the radiographic documentation is a little scarce, and the sources relating to the BPGP are not clearly cited in the bibliography. I would recommend improving these aspects before publishing.
Author Response
- “Dear Authors, I would like to congratulate you on your study. The modalities of the study are clearly illustrated, and the evaluationmethods are adequate. However the radiographic documentation is a little scarce, and the sources relating to the BPGP are not clearly cited in the bibliography. I would recommend improving these aspects before publishing.”
Reply: We did add new text into the final manuscript and bibliography addressing this comment and recommendation. A new case is added showing the amount of correction with the implant density recommended.
Round 2
Reviewer 1 Report
Dear Authors
I see that the current version has addressed some of the serious concerns raised in the previous round of review and I recommend your paper for publication